# Spatiotemporal distribution and sociodemographic and socioeconomic factors associated with primary and secondary syphilis in Guangdong, China, 2005–2017

**Shangqing Tang**[1☯], **Lishuo Shi**[2☯], **Wen Chen**[1], **Peizhen Zhao**[3,4], **Heping Zheng**[3,4], **Bin Yang**[3,4], **Cheng Wang**📷[3,4]*, **Li Ling**📷[1]*

**1** School of Public Health, Sun Yat-sen University, Guangzhou, Guangdong, China, **2** Clinical Research Center, The sixth affiliated hospital, Sun Yat-sen University, Guangzhou, Guangdong, China, **3** Dermatology Hospital, Southern Medical University, Guangzhou, Guangdong, China, **4** Institute for Global Health and Sexually Transmitted Disease, Southern Medical University, Guangzhou, Guangdong, China

☯ These authors contributed equally to this work.
* wangcheng090705@gmail.com (CW); lingli@mail.sysu.edu.cn (LL)

**Data Availability Statement:** The syphilis data analyzed in our study are not publicly available because of the limitation of data availability in the

## Abstract

### Background

Previous studies exploring the factors associated with the incidence of syphilis have mostly focused on individual-level factors. However, recent evidence has indicated that social-level factors, such as sociodemographic and socioeconomic factors, also affect the incidence of syphilis. Studies on the sociodemographic and socioeconomic factors associated with syphilis incidence are scarce, and they have rarely controlled for spatial effects, even though syphilis shows spatial autocorrelation.

### Methodology/Principal findings

Syphilis data from 21 cities in Guangdong province between 2005 and 2017 were provided by the National Notifiable Infectious Disease Reporting Information System. The incidence time series, incidence map, and space-time scanning data were used to visualize the spatio-temporal distribution. The spatial panel data model was then applied to explore the relationship between sociodemographic factors (population density, net migration rate, male: female ratio, and the number of health institutions per 1,000 residents), socioeconomic factors (gross domestic product per capita, the proportion of secondary/tertiary industry), and the incidence of primary and secondary syphilis after controlling for spatial effects. The incidence of syphilis increased slowly from 2005 (11.91 per 100,000) to 2011 (13.42 per 100,000) and then began to decrease, reaching 6.55 per 100,000 in 2017. High-risk clusters of syphilis tended to shift from developed areas to underdeveloped areas. An inverted U-shaped relationship was found between syphilis incidence and gross domestic product per capita. Moreover, syphilis incidence was significantly associated with population density ($\beta$ = 2.844, $P$ = 0.006), the number of health institutions per 1,000 residents ($\beta$ = -0.095, $P$ = 0.007), and the net migration rate ($\beta$ = -0.219, $P$ = 0.002).

data management rule of Guangdong Center for Skin Diseases and STI Control. Access to these data may be requested through the Guangdong Provincial Center for Skin Diseases and STI Control, China (contact via gdpfzx@vip.163.com) for researchers who meet the criteria for access to confidential data.

**Funding:** The research was funded by the National Natural Science Foundation of China to LL (82073664). The funders had no role in study design, data collection and analysis, decision to publish, or preparation of the manuscript.

**Competing interests:** The authors have declared that no competing interests exist.

## Conclusions/Significance

Our findings suggest that the incidence of primary and secondary syphilis first increase before decreasing as economic development increases further. These results emphasize the necessity to prevent syphilis in regions at the early stages of economic growth.

### Author summary

Syphilis is a sexually transmitted infection that continues to cause morbidity and mortality worldwide. The primary and secondary stages of syphilis are the most transmissive stages in the entire process of the disease. We analyzed primary and secondary (P&S) syphilis data from 2005 to 2017 in Guangzhou, China, provided by the National Notifiable Infectious Disease Reporting Information System. The results showed that the annual incidence rates of P&S syphilis slightly increased from 2005 to 2011 and then began to decrease in 2017. Cases of P&S syphilis were spatially clustered. The high-risk syphilis clusters tended to shift from developed areas to underdeveloped areas. There may be an inverted U-shaped relationship between the level of economic development and the incidence of P&S syphilis, suggesting that the incidence of P&S syphilis first increased before decreasing as the level of economic development increased further. These results emphasize the necessity of preventing syphilis at locations in the early stage of economic growth. Investments in syphilis prevention education for people in regions at early development stages may mitigate the increasing cost of syphilis to future healthcare systems.

## Introduction

Syphilis is a bacterial infectious disease caused by *Treponema pallidum*. It continues to cause morbidity and mortality worldwide through sexual and vertical transmission [1–2]. According to the World Health Organization, 6.3 million new cases were reported globally each year [3], totaling approximately 43 million cases globally [4], resulting in more than 107,000 deaths [5]. In China, from 2005 to 2014, the incidence of syphilis increased more than any other notifiable infectious disease [6].

Syphilis is most transmissive during the primary and secondary (P&S) stages of its progression [7]. Early prevention of P&S syphilis can effectively stop its transmission [8]. Identifying areas with a high risk of P&S syphilis can provide recommendations for its early prevention, so that health services and resources can be more efficiently distributed [9]. Exploring the sociodemographic and socioeconomic factors associated with the incidence of P&S syphilis can help identify the characteristics of high-risk areas and facilitate the development of effective responses [10].

Previous studies have shown that the incidence of P&S syphilis is affected by certain sociodemographic factors [11–12]. The size and structure of a population and the health care resources per capita are closely related to syphilis incidence. Areas with a large population, unbalanced sex ratios, and fewer health resources tend to have a a higher incidence of P&S syphilis [13–14].

Regarding socioeconomic factors, studies presented conflicting results concerning incidence. Research conducted in high-income regions, such as the United States and Europe, have shown that syphilis is more prevalent in poor areas [12,15]. However, contradictory conclusions have been reported in studies from middle-income and low-income countries, including China, indicating that more prosperous areas have a higher risk of syphilis [16–17]. Data

from research on other diseases suggest that the explanation for this contradiction may be that there is an inverted U-shaped relationship between economic development and syphilis incidence [18–19]. However, research was lacking to support this claim.

Furthermore, most previous studies on the sociodemographic and socioeconomic factors associated with P&S syphilis have not evaluated the effects of spatial autocorrelation, but have regarded each region as an independent geographical unit [20]. Cases of P&S syphilis have been shown to be spatially clustered [21], and therefore, the results may be biased if spatial autocorrelation is not considered in the analysis [22].

Thus, the aim of this study was to describe the spatial and temporal distribution of P&S syphilis in Guangdong province from 2005 to 2017, and to investigate the sociodemographic and socioeconomic factors associated with the incidence of P&S syphilis, while controlling for the spatial effect.

## Methods

### Study area

Guangdong is a coastal province in South China (Fig 1) that reported the highest number of syphilis cases (55,777) of all the provinces in China in 2017 [23]. It consists of 21 municipal-level cities. From 2005 to 2017, the internal economic conditions and population composition were highly heterogeneous between cities of Guangdong province [17,24], which was conducive to providing sufficient data for our study. For example, cities in the south-central area of Guangdong, which is known as the Pearl River Delta region, were densely populated and economically developed. Nine cities in this region accounted for 80% of the gross domestic product (GDP) of Guangdong province, but made up less than 30% of the land area of the province [25]. Another 12 cities accounted for only 20% of the GDP of the province.

### Data source

Case-based P&S syphilis data from 2005 to 2017 were obtained from the web-based National Notifiable Infectious Disease Reporting Information System. Every medical institution at the county level or above in China was obliged to report all diagnosed cases of P&S syphilis online within 24 hours after the diagnosis. The data were audited by the Centers for Disease Control and Prevention to ensure authenticity and reliability. The number of P&S syphilis cases was determined at the municipal level within Guangdong province. We used the annual P&S notification rate (per 100,000 population) for all 21 cities in Guangdong province as an outcome variable in the spatiotemporal analyses. As no private patient data were included in the study, approval from an ethics committee was not required.

In line with previous studies, four sociodemographic variables were selected, namely, population density [13], net migration rate [26], male:female ratio [14], and the number of health institutions per 1,000 residents [13]. The population density was defined as the number of people per square kilometer [27]. The net migration rate was calculated as the ratio of the non-registered population to the total resident population, expressed as a percentage [28]. Socioeconomic variables included GDP per capita [17] and the proportion of secondary/tertiary industry [29] to represent the economic level and structure, respectively. All data were derived from the *Guangdong Statistical Yearbook (2006–2018)* and the *Guangdong Health and Family Planning Statistical Yearbook (2005–2017)*.

In addition, considering that meteorological factors may affect the incidence of syphilis [30], we incorporated the following municipal-level meteorological factors into the model as control variables: annual average temperature (˚C), average relative humidity (%), precipitation (mm), and number of daylight hours (h). Meteorological data were obtained from the

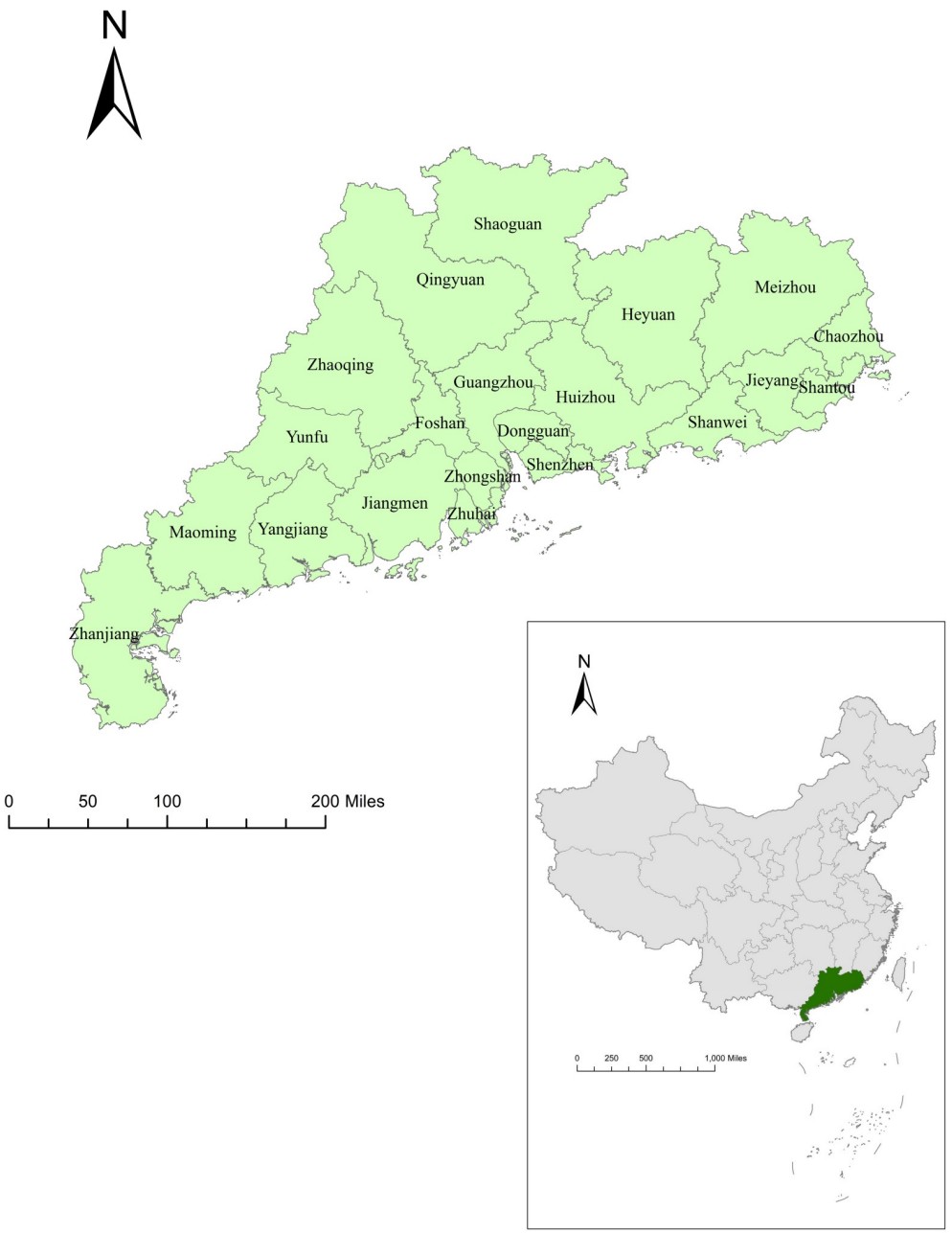

**Fig 1. Location of Guangdong province in China.** Base layers of the maps were downloaded from Resource and Environment Science and Data Center (http://www.resdc.cn/data.aspx?DATAID=201).

China Meteorological Data Sharing Service System (http://data.cma.cn), which is publicly accessible. The meteorological data of a city was calculated by taking the annual average value of all meteorological station data of each city.

## Statistical analysis

**Description of the temporal and spatial distribution.** To visualize the temporal and spatial variations in the incidence of P&S syphilis in Guangdong province from 2005 to 2017, a

time curve of the monthly incidence rate was plotted and choropleth maps were constructed for every two-year period.

**Space-time scan analysis.**   Kulldorff's retrospective space-time scan analysis, based on the discrete Poisson model [31], was used to identify spatio-temporal clusters of P&S syphilis at the municipal level, using SaTScan V-9.6 software (https://www.satscan.org/). The principle of scanning statistics is to build a scanning window with the geographic area as the base and the time as the height [32]. The center position of the window and the radius of the base were varied repeatedly, and a statistic was generated to test the difference between the actual number of patients and the theoretical value in the window. This algorithm was able to identify the years during which syphilis cases were clustered in Guangdong province, and the location of the clusters. The significance of the clusters was deduced based on Monte Carlo simulations. If the null hypothesis was rejected, the likelihood ratio of the scan window area was statistically significant, indicating that there was a cluster during this period. The maximum size of the scanning window was set as 50% of the total population at risk in our study. The time aggregation scan length was set to 1 year, so that we could observe changes in the cluster every year.

**Spatial panel data model.**   Before building the model, Moran's I test was used to determine whether the data were suitable for the spatial panel data model [33]. The two most commonly used spatial panel data models are the spatial lag model (SLM) and the spatial error model (SEM) [34]. The SLM takes the influence of neighboring units' dependent variables into account, while the SEM reflects the influence of the neighboring units' non-observable components. The results of the Lagrange multiplier (LM) and robust LM tests indicated that the SLM was more appropriate than the SEM for interpreting our data [35]. The formula for the SLM was as follows:

$$y_{it} = \mu_i + \gamma_t + \rho\sum_{j=1}^{N} W_{ij}y_{jt} + X_{it}\beta + \varepsilon_{it}, \tag{1}$$

where $i$ and $t$ represent different units and different time points, respectively; $y_{it}$ is the dependent variable of unit $i$ at time $t$ and $\mu_i$ and $\gamma_t$ are the spatial specific effect and temporal specific effect, respectively; $\rho$ is the spatial autoregressive coefficient that reflects the degree of spatial interaction; $W_{ij}$ is a weight matrix used to express the spatial relationship of cities; $X_{it}$ is a set of independent variables; $\beta$ is the regression coefficient; and $\varepsilon_{it}$ is the random error [36].

To test the hypothesis of an inverted U-shaped relationship between economic level and P&S syphilis incidence, GDP per capita and its squared term were both included in the model. Log transformation was used to reduce the overdispersion of some non-normally distributed data before the model analysis. The tests and models were completed using Matlab R2019a (Mathworks Inc., Natick, MA, USA).

## Results

### Description of the temporal and spatial distributions

From January 2005 to December 2017, Guangdong province reported 147,662 P&S syphilis cases. The number of cases in each month ranged from 416 to 1,392 (Fig 2). The annual incidence rate showed a slight increase from 2005 (11.91 cases per 100,000 population) to 2011 (13.42 cases per 100,000 population) and then began to decrease, reaching 6.55 cases per 100,000 population in 2017. P&S syphilis was most commonly diagnosed from June to October every year, which indicated a seasonal periodicity.

Fig 3 shows a plot of the disease distribution maps, which demonstrates the spatial heterogeneity of P&S syphilis incidence. At the beginning of the period studied, the high-risk areas were predominantly concentrated in the northern region and the economically developed

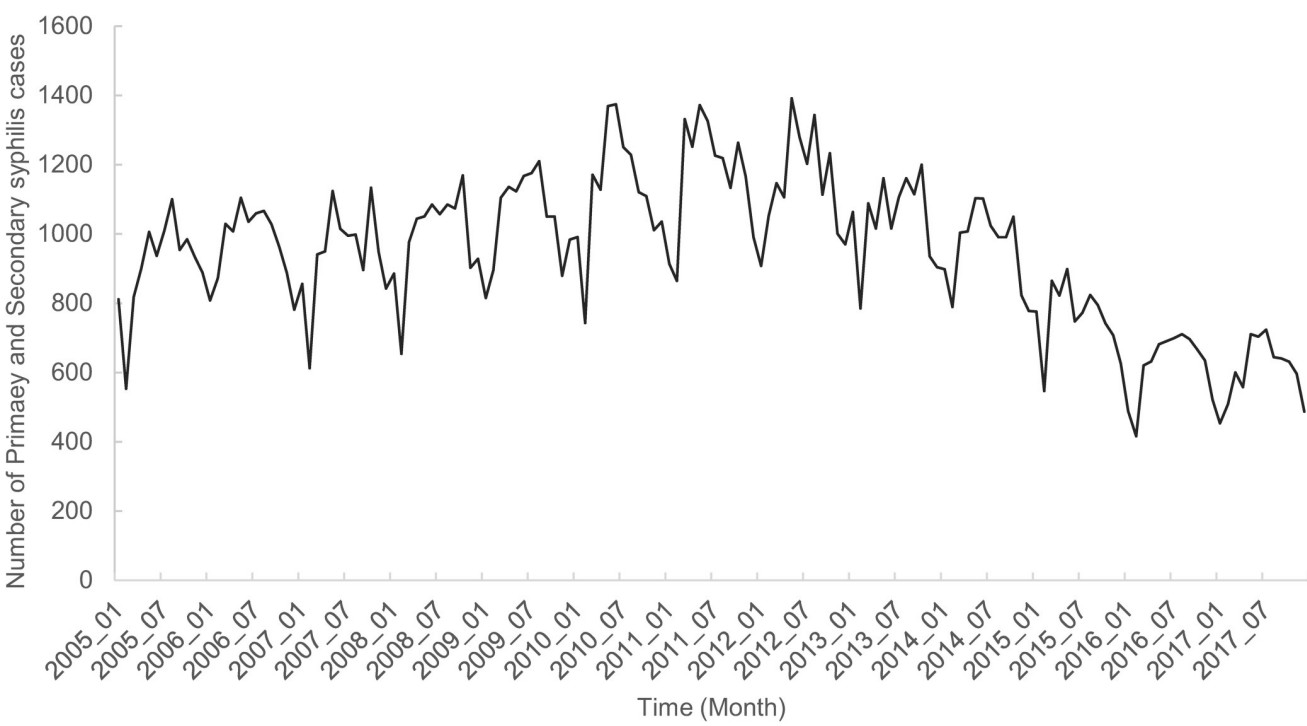

**Fig 2. Number of P&S syphilis cases in Guangdong province from 2005 to 2017.**

south-central regions, such as the cities of Guangzhou, Shenzhen, Zhuhai, Foshan, and Zhongshan. As time progressed, the incidence of P&S syphilis in these regions showed a decreasing trend. The rate of decrease was greater in the northern region than in the south-central region. Cities in the eastern and western regions of the province, which are relatively economically underdeveloped, showed an initial increase in the incidence of P&S syphilis during 2005–2013 and then a decrease. After 2013, the upward trend in P&S syphilis incidence in most cities of Guangdong was brought under control.

## Spatial cluster identification

Two significant clusters were identified by the space-time scan analysis (Fig 4). The most likely cluster from 2005 to 2010 ($P < 0.001$), indicated in green, remained in the south-central area, which includes the cities of Guangzhou, Shenzhen, Foshan, Dongguan, and Zhongshan. This was the most economically developed area of Guangdong province during that time. The center of the cluster was 23.35 N and 113.54 E, with a radius of 97.66 km. From 2011 to 2014, a secondary cluster ($P < 0.001$), indicated in blue, was identified in the northeast region of Guangdong. This cluster consisted of three less-developed cities in Guangdong, namely, Huizhou, Meizhou, and Heyuan city. The center of the cluster was 24.04 N and 114.96 E, with a radius of 115.05 km.

## Sociodemographic and socioeconomic factors associated with P&S syphilis incidence

The results of Moran's I test indicated that the prerequisite for the model was satisfied. The LM and robust LM statistics of the SLM were more significant than those of the SEM (Table 1). The spatial lag panel data model was used to analyze the sociodemographic and socioeconomic factors associated with P&S syphilis incidence.

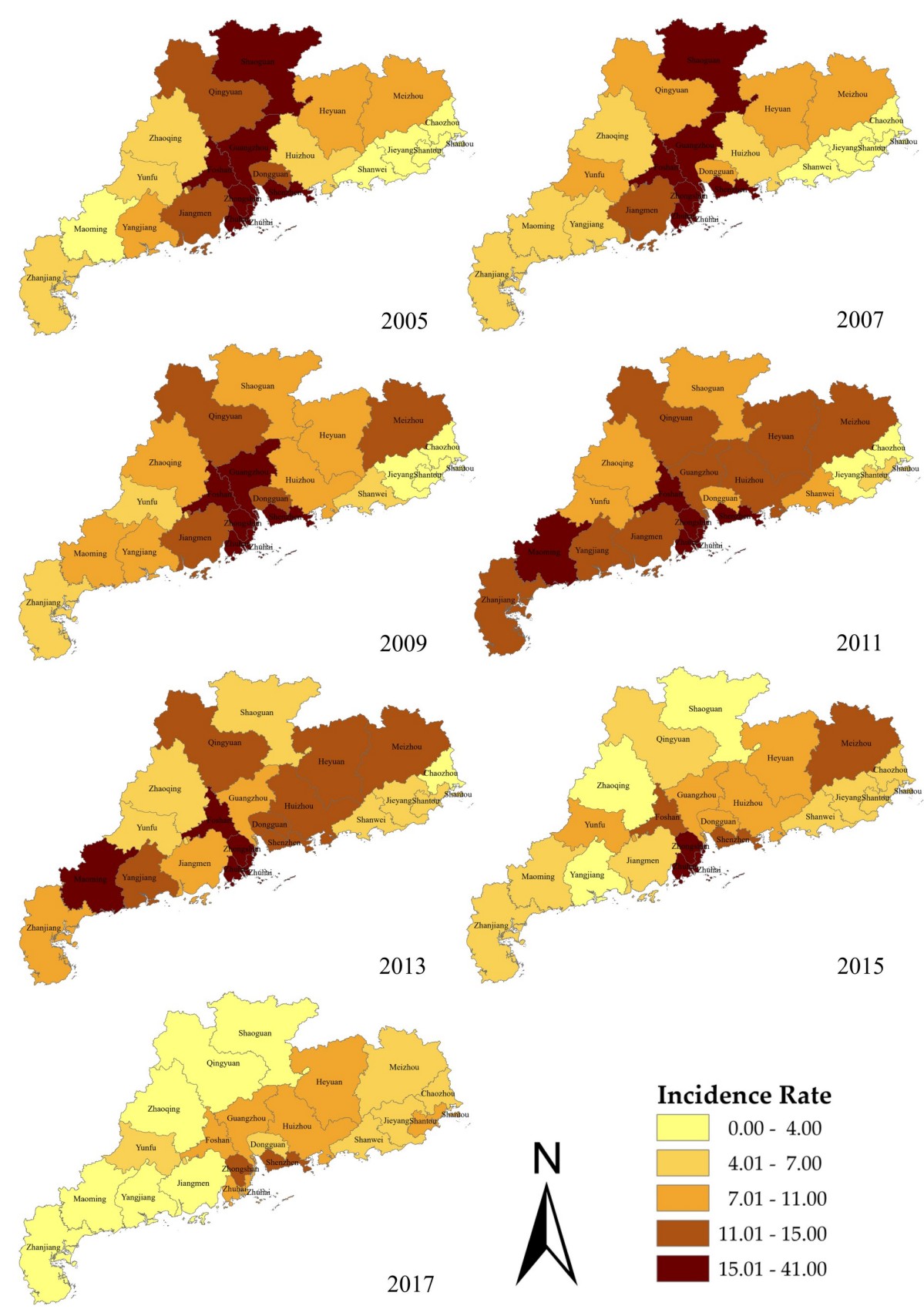

**Fig 3. Spatial distributions of the incidence rates (per 100,000 population) of P&S syphilis.** Base layers of the maps were downloaded from Resource and Environment Science and Data Center (http://www.resdc.cn/data.aspx?DATAID=201).

The spatial autoregression coefficient ($\rho$) of the spatial panel data model was statistically significant, indicating the existence of neighborhood effects (Table 2). Thus, the incidence of P&S syphilis in a city was positively affected by the incidences of P&S syphilis in neighboring cities. These results demonstrated that an inverted U-shaped relationship existed between P&S syphilis incidence and GDP per capita, as indicated by the positive value of the GDP per capita coefficient and the negative value of the GDP per capita squared coefficient. We plotted a scatter diagram of GDP per capita and the incidence rates of P&S syphilis to graphically present the inverted U-shaped relationship (S1 Fig). In terms of sociodemographic variables, population density was positively associated with P&S syphilis incidence, while the net migration rate and the number of health institutions per 1,000 residents were negatively associated with P&S syphilis incidence. The distribution of the above variables during 2005–2017 was shown in S2–S5 Figs. The other factors tested had no statistically significant effect.

## Discussion

Our study is the first to analyze the relationship between sociodemographic and socioeconomic factors and P&S syphilis incidence, while taking the spatial autocorrelation into consideration. We found that there was an inverted U-shaped relationship, rather than a linear relationship, between P&S syphilis incidence and GDP per capita. This stresses the importance of implementing early prevention strategies for syphilis in regions at the early stages of economic growth.

In terms of temporal distribution, P&S syphilis incidence showed significant periodicity and seasonality. The high-occurrence period was from June to October, which is similar to the results reported for other cities and countries [37–38]. As for the spatial distribution, around

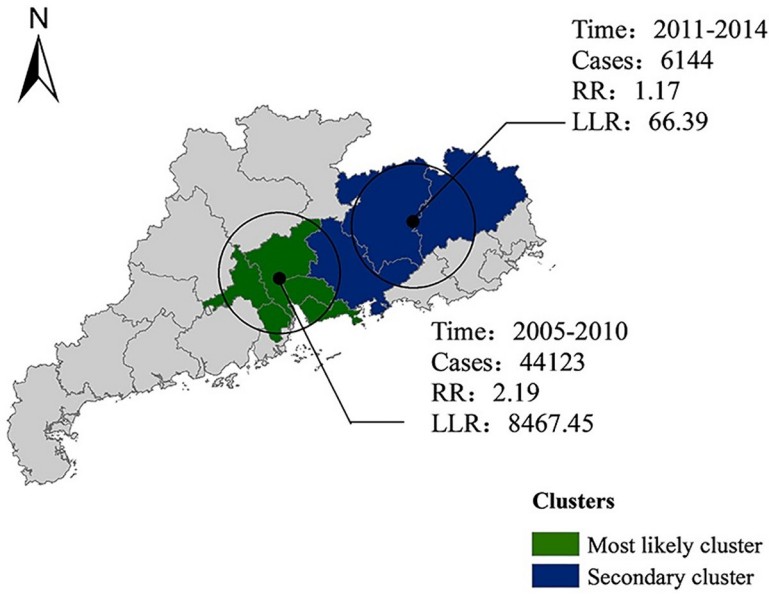

**Fig 4. The spatial cluster of P&S syphilis in Guangdong province from 2005 to 2017.** Base layer of the map was downloaded from Resource and Environment Science and Data Center (http://www.resdc.cn/data.aspx?DATAID=201).

**Table 1. Results of the Moran's I test and LM tests of the model.**

| Type of test | Moran's I test | LM tests | | | |
|---|---|---|---|---|---|
| | | LM lag test | LM error test | Robust LM lag test | Robust LM error test |
| Test statistic value | 5.953 | 35.470 | 29.121 | 7.504 | 1.154 |
| P value | < 0.001 | < 0.001 | < 0.001 | 0.006 | 0.283 |

LM, Lagrange multiplier

the year 2011, the high-incidence clusters tended to shift from the prosperous coastal area to the surrounding inland cities, which were relatively poorer. A similar situation was observed for the entire country, suggesting an association between the syphilis epidemic and economic trends [39].

After adjusting for the effects of spatial autocorrelation, our results showed that the trend in P&S syphilis incidence with the increase in economic development was divided into two stages. At the early stage of economic development, the incidence of syphilis showed an upward trend. This may be due to the rapid development of the commercial sex industry and an openness in people's sexual beliefs after China implemented more open and liberal economic policies [40–41]. Commercial sexual behavior and high-risk sexual behavior increased during this period. It has been reported that 14% of sex workers in Guangdong province have syphilis, with more than 50% of them engaging in unprotected sex with their customers [42], which may contribute to the spread of syphilis. However, the decreasing P&S syphilis incidence could be expected when the economic development has reached a relatively high level. With increased economic development, public security management capabilities are strengthened to effectively control illegal prostitution and improve people's health education and awareness [43]. At the same time, similar to the Kuznets' curve theory [44], sufficiently affluent areas are able to provide comprehensive coverage of health services and thus more equitable access to syphilis screening and treatment services [39]. For example, the screening rate for pregnant women in Shenzhen, the wealthiest city in Guangdong province, increased from 89.8% to 97.2% from 2002 to 2012, which provided an opportunity for the early detection and treatment of syphilis.

**Table 2. Results of the spatial panel data model.**

| Variables | Coefficient | 95% Confidence Interval | t-value | P value |
|---|---|---|---|---|
| **Sociodemographic factors** | | | | |
| Log (population density) (number of people/km$^2$) | 2.844 | 0.944 to 4.744 | 2.768 | 0.006 |
| Net migration rate (%) | -0.219 | -0.350 to -0.088 | -3.127 | 0.002 |
| Male:female ratio | 3.168 | -2.673 to 9.009 | 1.020 | 0.308 |
| Number of health institutions per 1,000 residents | -0.095 | -0.161 to -0.029 | -2.690 | 0.007 |
| **Socioeconomic factors** | | | | |
| Log (GDP per capita) (RMB) | 5.747 | 2.557 to 8.937 | 3.413 | <0.001 |
| Log (GDP per capita)$^2$ | -0.246 | -0.408 to -0.084 | -2.875 | 0.004 |
| Proportion of secondary industry (%) | -0.004 | -0.022 to 0.014 | -0.395 | 0.693 |
| Proportion of tertiary industry (%) | -0.015 | -0.034 to 0.004 | -1.424 | 0.154 |
| Spatial autoregressive coefficient ($\rho$) | 0.363 | 0.244 to 0.482 | 5.872 | <0.001 |
| $R^2$ | 0.782 | | | |

*Note*: Meteorological factors, including average temperature, average relative humidity, precipitation, and daylight hours were controlled in the model.

Consistent with previous studies [45], the results of this study suggest that areas with a high population density had higher incidences of P&S syphilis, as the chance of exposure was increased. The statistical relationship between P&S syphilis incidence, net migration rate, and the number of medical institutions may have resulted from a chain reaction from economic growth. Economic growth results in the allocation of more resources to healthcare [46]. Meanwhile, considering the push and pull theory, the economic and medical resources were the pull factors of migration [47]. People tended to migrate to prosperous places where there were more health institutions and greater attention to people's health.

Our results provided new insights into the relationship between economic factors and syphilis incidence. They emphasize the necessity of preventing syphilis and other infectious diseases with similar transmission routes in susceptible populations while ensuring sustainable economic development. Regions at the early stages of economic growth might have experienced a period when the risk of syphilis continues to increase, which means that in the foreseeable future, further economic development will be accompanied by an increase in the number of syphilis cases if timely interventions are not implemented. Early prevention measures such as the education of high-risk groups and the popularization of syphilis prevention strategies [48] may be implemented in these areas. Investing in the health education of people in regions at the early stages of economic development may mitigate the increase in future healthcare costs. For areas with greater economic development, syphilis testing needs to be further modernized and universalized. Priorities for syphilis prevention may include the improvement of healthcare systems to ensure that all residents have equal access to timely detection and treatment when necessary.

This study has some limitations. First, underreporting is a common problem in surveillance data, especially for sexually transmitted diseases, which may breach patients' privacy. However, the surveillance case-based reporting data used in this study were the most complete dataset currently available. Second, due to the availability of data, the units of the panel data model were cities and years and it was difficult to analyze smaller units. Therefore, some variation in the data, such as seasonal changes, may not have been detected. Finally, our study showed only a correlation, not a causal relationship. The determination of causality needs to be inferred through logical judgment and further practical investigation, which may cause some difficulties when formulating targeted prevention policies.

## Conclusions

The findings of this study provide new empirical evidence for the sociodemographic and socioeconomic factors associated with P&S syphilis incidence. Residents of cities in the early stages of economic development may have an increased risk of syphilis. Early prevention strategies, such as health education for high-risk groups or even the entire population, should be implemented in these locations as early as possible. The results of our study also reiterate the important role that economic development plays in improving health outcomes, and particularly, its role in curbing the epidemic of syphilis and other similar infectious diseases. When the economy develops to a certain level, the incidence of syphilis is expected to gradually decrease.

## Supporting information

**S1 Fig. The scatter plot of GDP per capital and incidence rates of P&S syphilis.**
(TIFF)

**S2 Fig. Spatial distributions of population density.** Base layers of the maps were downloaded from Resource and Environment Science and Data Center (http://www.resdc.cn/data.aspx?DATAID=201).
(TIF)

**S3 Fig. Spatial distributions of net migration rate.** Base layers of the maps were downloaded from Resource and Environment Science and Data Center (http://www.resdc.cn/data.aspx?DATAID=201).
(TIF)

**S4 Fig. Spatial distributions of number of health institutions per 1,000 residents.** Base layers of the maps were downloaded from Resource and Environment Science and Data Center (http://www.resdc.cn/data.aspx?DATAID=201).
(TIF)

**S5 Fig. Spatial distributions of GDP per capita.** Base layers of the maps were downloaded from Resource and Environment Science and Data Center (http://www.resdc.cn/data.aspx?DATAID=201).
(TIF)

## Acknowledgments

We thank the Guangdong Provincial Center for Skin Diseases & Sexually Transmitted Infections Control for providing the data.

## Author Contributions

**Conceptualization:** Shangqing Tang, Lishuo Shi, Li Ling.

**Data curation:** Bin Yang, Cheng Wang.

**Formal analysis:** Lishuo Shi.

**Funding acquisition:** Heping Zheng, Cheng Wang, Li Ling.

**Investigation:** Peizhen Zhao, Heping Zheng.

**Methodology:** Shangqing Tang, Lishuo Shi, Wen Chen.

**Project administration:** Cheng Wang.

**Resources:** Bin Yang.

**Supervision:** Cheng Wang, Li Ling.

**Validation:** Shangqing Tang, Peizhen Zhao.

**Writing – original draft:** Shangqing Tang.

**Writing – review & editing:** Shangqing Tang, Lishuo Shi, Wen Chen, Li Ling.

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
