## [Decision Letter · Decision Letter 0]

24 Apr 2021

Dear Dr. Ling,

Thank you very much for submitting your manuscript "Spatiotemporal distribution and sociodemographic, socioeconomic associated factors of primary and secondary syphilis in Guangdong, China, 2005-2017" for consideration at PLOS Neglected Tropical Diseases. As with all papers reviewed by the journal, your manuscript was reviewed by members of the editorial board and by several independent reviewers. The reviewers appreciated the attention to an important topic. Based on the reviews, we are likely to accept this manuscript for publication, providing that you modify the manuscript according to the review recommendations. 

Thank you for submitting this manuscript to PLoS Neglected Tropical Diseases which has now completed the review process. The reviews are favourable but specific changes to the manuscript have been suggested. In addition to responding to these, if you intend to resubmit a revised manuscript please add to the details in the methods on the source of the primary and secondary syphilis data - are they publicly available? is there a web link? and if not how would other researchers obtain these data?

Sincerely,

Graham P Taylor, MB, DSc

Associate Editor

Mathieu Picardeau

Deputy Editor

Thank you for submitting this manuscript to PLoS Neglected Tropical Diseases which has now completed the review process. The reviews are favourable but specific changes to the manuscript have been suggested. In addition to responding to these, if you intend to resubmit a revised manuscript please add to the details in the methods on the source of the primary and secondary syphilis data - are they publicly available? is there a web link? and if not how would other researchers obtain these data?

Reviewer's Responses to Questions

**Key Review Criteria Required for Acceptance?**

**Methods**

-Are the objectives of the study clearly articulated with a clear testable hypothesis stated?

-Is the study design appropriate to address the stated objectives?

-Is the population clearly described and appropriate for the hypothesis being tested?

-Is the sample size sufficient to ensure adequate power to address the hypothesis being tested?

-Were correct statistical analysis used to support conclusions?

-Are there concerns about ethical or regulatory requirements being met?

Reviewer #1: The objectives the study are clearly written, study design appropriate, sample size is ok for the study.

The statistical analysis done supports the conclusion.

Reviewer #2: The objectives of this study were clearly presented and easily tested. The study design appears appropriate to the hypotheses tested. This is a relatively simple and straightforward study that provides needed insight regarding the sociodemographic and socioeconomic factors surrounding P&S Syphilis transmission. 

Line 121: Can you provide a map of the region both broad and fine scale (inset) to help the reader with the area?

Line 124: Start the sentence with, “In 2017, there was great…” and do not switch between tenses

Line 128: Check journal standards, but values less than 10 are spelled out, i.e., ‘nine’ 

Line 134: Does this surveillance system have an official name?

General: How often are cases reported? And what is the spatial resolution of those reports (to which level of governance are they aggregated?)

Line 137: “will be” or “was”? Check your use of tense throughout

Line 141: Do you mean ‘number of health institutions’?

Line 142: “person counts”? ‘number of people per square kilometer’ ?

Line 145: Add a comma after ‘structure’

Line 152: Replace ‘sunshine” with ‘daylight’

Lines 151-152: Provide the units in which these metrics were recorded. Were these data aggregated to the administrative units? If so, how?

**Results**

-Does the analysis presented match the analysis plan?

-Are the results clearly and completely presented?

-Are the figures (Tables, Images) of sufficient quality for clarity?

Reviewer #1: The analysis presented match the analysis plan.

results clearly and completely presented

 figures (Tables, Images) are of sufficient quality

Reviewer #2: Lines 203-205: “A peak period”? Is there a peak monthly period each year? Or were these data aggregated?

Line 207: Regarding the time series maps. It appears that the incidence rate hot spot moves from north to south regions, until the rate nearly dissolves. What is this pattern I’m seeing? 

Line 208: The term ‘hot spot’ is used here, but it has a specific meaning in spatial analyses that were not conducted in this study. It would be best to find a different term.

Line 225: There is an unnecessary period after 1.17

Line 237: Change ‘autoregressive’ to ‘autoregression’

Line 254: Add ‘factors’ after sociodemographic

Spatial clusters identification: Does the algorithm determine grouping of years, or was this done by the authors?

Results: Avoid redundancy by either 1) putting your statistics in a table OR 2) adding them to the body of the text. Generally, you should not do both. Example Line 230 

Results: I recommend that you create a map (or series of maps) that shows distribution of your variables (e.g., population density, net migration, health institutions, etc.). It would be very helpful to visualize these data spatially. It could just be an appendix or supplemental figure.

Results: Can you graphically present the U-shaped relationships you found in this study?

**Conclusions**

-Are the conclusions supported by the data presented?

-Are the limitations of analysis clearly described?

-Do the authors discuss how these data can be helpful to advance our understanding of the topic under study?

-Is public health relevance addressed?

Reviewer #1: The data presented supports the conclusion

Reviewer #2: The conclusion are supported by the results and data.

**Editorial and Data Presentation Modifications?**

Reviewer #1: Accept

Reviewer #2: There is quite a bit of tense-switching throughout the manuscript (past, present, and future tenses were all noted). Please check thoroughly.

**Summary and General Comments**

Reviewer #1: (No Response)

Reviewer #2: Overall, this was simple and to the point. My recommendations are minor but strongly suggested.

PLOS authors have the option to publish the peer review history of their article (what does this mean?). If published, this will include your full peer review and any attached files.

Reviewer #1: No

Reviewer #2: No

Figure Files:

Data Requirements:

Reproducibility:

References

---

## [Editor Report · Decision Letter 1]

2 Jul 2021

Dear Dr. Ling,

We are pleased to inform you that your manuscript 'Spatiotemporal distribution and sociodemographic and socioeconomic factors associated with primary and secondary syphilis in Guangdong, China, 2005-2017' has been provisionally accepted for publication in PLOS Neglected Tropical Diseases.

Best regards,

Graham P Taylor, MB, DSc

Associate Editor

Mathieu Picardeau

Deputy Editor

---

## [Editor Report · Acceptance letter]

26 Jul 2021

Dear Dr. Ling,

We are delighted to inform you that your manuscript, "Spatiotemporal distribution and sociodemographic and socioeconomic factors associated with primary and secondary syphilis in Guangdong, China, 2005-2017," has been formally accepted for publication in PLOS Neglected Tropical Diseases.

Best regards,

Shaden Kamhawi

co-Editor-in-Chief

Paul Brindley

co-Editor-in-Chief
